# Sustainable Enterprise Capital Management

**Dariusz Klimek** 

Faculty of Management and Production Engineering, Lodz University of Technology, 90-924 Lodz, Poland; dariusz.klimek@p.lodz.pl; Tel.: +48-42-6313766

**Abstract:** The paper presents the idea of a new method of sustainable enterprise capital management. The idea is based on a principle that states that the faster an enterprise achieves and maintains a balance between its capitals, the more effective the management is. The concept forms a part of the search for an alternative to net profit, which is a basic but outdated measure in modern times. After the introduction, the author outlines the basic principles and foundations of the new method. After introducing the principles and assumptions of the new idea, the author, in his paper, describes the studies that aimed at a practical checking of the possibilities of measuring the effectiveness of the company's economic condition and the effects of management in accordance with the new principles. They were carried out in one of the food industry companies, whose shares are listed on the Warsaw Stock Exchange. Two new economic indicators were used in the study. The general conclusion of the study is as follows: new principles of sustainable capital management can be applied in practice to measure the effectiveness of the enterprise and the work of the management board, but there are still many conditions and problems listed in the paper that should be the subject of further work. The issue that still needs a lot of research within the whole idea is the problem of valuing the current and optimal level of corporate capital. The conclusions outline the strengths and weaknesses of the new method.

**Keywords:** enterprise resources; sustainable management; enterprise capital; management efficiency

## 1. Introduction

An analysis of subject-matter literature and opinions clearly indicates that theoreticians and practitioners of management are convinced that a new paradigm of measuring enterprise efficiency is needed, one that would be an alternative to profit and would treat this task as a continuous evolutionary process. The search for such a measure of effectiveness has been going on for several decades (Fisher and McGowan 1983; Eccles 1991; Crowther 1996). In the second half of the last century, despite the fact that topics such as intellectual and human capital or corporate social responsibility have already emerged in social and scientific discussion, they went in the same direction as profit, i.e., maximizing or optimizing it. The ones worth mentioning are: E. Penrose's growth theory, sales maximization; the Baumola model, maximizing usability for managers by increasing discretionary expenses; the Williamson model, maximizing growth; the Marris model. A little later, as measures of the company's achievements, alternative to the traditional accounting profit, there were also attempts to popularize the so-called economic value added (EVA), where the basis is the rate of return on investment (ROI) and the subject of measurement is the corrected enterprise value and a measure in the form of free cash flows (FCF) in which the basis is pure cash flow and the subject of measurement is the value of the enterprise (Ratnatunga 2002).

However, the truth is that after several decades of searching, profit is still the basic economic measure. However, such situation cannot last forever and critical voices often appear in the literature (Coates et al. 1995; Crowther et al. 1998). This is because of the following two disadvantages of net

profit, i.e., an option often used in business practice: the instrumental use of profit by managers and the defective design of this measure.

An example of an instrumental use of profit can be an international company within the holding structure, which can easily bring profits to a country with lower taxes by, e.g., maximizing the remuneration of suppliers from another holding company located abroad. If it is an employees' company, the owners may want to maximize wages at the expense of profit if a domestic family-owned company can have goals beyond profit. If the remuneration for the members of the management board largely depends on profit, then this profit can be achieved without caring for the long-term development of the company, which may involve the need for higher costs and lower profit in a given period, which could reduce the board's remuneration. If these are municipal companies, they must plan the profit so that it is not too high, as they will be accused of fixing prices for their services rendered for residents on an excessive level, or too low, because it may indicate bad management.

The design of the profit meter does not correspond to modern times. Firstly, it is not entirely clear whether profit is a measure of the economic condition of an enterprise or a measure of the ability and effort of a manager and employees. The internal condition and the external conditions of entities are usually different. Sometimes, improving the situation within a company requires many years of hard work, and in some cases it may even be impossible. This process also involves costs that reduce results. For another company, there is not much work and its situation is very good and stable. Big profits can easily be made. Secondly, it also has no properties to assess the efforts and expenditure of a given period for the results of subsequent years. This results in the managers' attention being focused on short-term (e.g., annual) financial results, at the expense of future investments or investments not transferring directly to revenues. This can be particularly noticeable when a manager knows that he/she will not necessarily be employed in the following years. Thirdly, this measure is based on financial data from the accounting system, which is created primarily for tax purposes, and does not include the real assets of entities, e.g., human or social capital.

Generally, a situation where profit as a basic measure is permissible with unlimited resources or the fact of not taking into account the resources in order to achieve the planned goals, is not desirable nowadays when attention is paid to natural and, e.g., human resources, which can also be limited, difficult to access, and, above all else, expensive. The article is part of the search for an alternative to profit as the current basic efficiency measure in the world. It is important primarily for company managers and owners who can better use the company's capital and achieve better business results as a result of increasing revenues or reducing costs. However, it should also be emphasized that this way of measuring efficiency pays special attention to human and social capital, which is often underestimated when using profit as a measure of effectiveness. In addition, it includes elements of corporate social responsibility in the assessment, combining public utilities with commercialism.

The sustainable concept was based on the development of many scientific theories, in particular the theory of resources, referring to the approach of Schumpeter and the achievements of Penrose, known in the literature as the Resource-Based View of the Firm (RBV) (Schumpeter 1934; Penrose 1955, 1956, 1960; Barney 1991; Hall 1992; Peteraf 1993; Kor and Mahoney 2000; Kor et al. 2007, 2016). No less important to the concept was the development of the theory of intellectual capital, particularly the definition of social capital and determining its impact on other capitals (Nahapiet and Ghoshal 1998; Stewart and Ruckdeschel 1998; Leana and Van Burren 1999; Lin 2001; Adler and Kwon 2002). Finally, the work of researchers, related to the company's capital valuation, was important for the empirical part of the concept (Bontis 2002; Rodov and Leliaert 2002; Tan et al. 2007; Jolanta 2008; Maditinos et al. 2011; Jędrych and Klimek 2018).

Publications on sustainable development started to appear in the literature for several decades and, in the first periods, they primarily concerned issues related to ecology and natural resources. The economic aspect, which was both social and cultural in the context of sustainable development, did not appear until several years (Abri et al. 2017; Ike et al. 2019; Nikolaou et al. 2019; Popescu and Popescu 2019; Popescu 2019; Schönborn et al. 2019; Usar et al. 2019).

Within recent years, several methods of sustainable resource management at the company level have appeared in publications (Ifko 2016; Capatina et al. 2017; Kosacka and Werner-Lewandowska 2017; Oh et al. 2018; Schrippe and Ribeiro 2018; Ashrafi et al. 2019). Academics have also been trying to improve the methods for valuing the individual capital of enterprises (Meszek 2015), as well as the methods for creating tools for capital management (Gogan et al. 2014; Łataś and Walasek 2016). Certain academics have paid a lot of attention to the studies on managing the sustainable development of selected capitals within the realm of macroeconomic instability (Absalyamova et al. 2015).

## 2. Methodology

The purpose of the research described in the article was to empirically check the assumptions of the new way of measuring the economic efficiency of the enterprise and the effects of management board activities. The new concept assumes that enterprise management is a continuous process of balancing the enterprise's capital level in order to achieve its maximum efficiency and the optimal use of resources. To simplify this, the point is to find a balance point between several of the capital constituting the enterprise.

During research, the author tries to answer the following research question: do the indicators developed by the author measure the effectiveness of the enterprise in practice?

Studies on the impact of capital on the results of enterprises, particularly the impact of intellectual capital, including human capital, on an enterprise has, already had a long history (Griffith and Harvey 2004; Gottwald et al. 2015; Hashima et al. 2015; Gogan et al. 2016; Khan and Quaddus 2018).

The principles of managing the company's balanced capital, in a simplified way, can be described in several points:

(1) The goal of business management is to obtain the best level of efficiency, understood as the maximum effect from specific resources. In practice, this goal is to reach a capital equilibrium point, the sooner a manager manages to achieve a relative balance and maintain it, the more effective the company is. The balance between the company's capitals should in no case be equated with equal monetary value, the balance usually occurs between the capital of different monetary value.

(2) The monetary values of capitals are subject to constant change. Hence, in practice, management consists of increasing or decreasing the level of a given capital by adjusting its level to the level of other capital or by adjusting the level of other capital to changes that occurred in one of them. There may also be a situation when the increase in capital (s) does not make it necessary to react with other capital(s), as their level was already higher before.

(3) Capitals interact with each other regardless of any actions taken by managers. An increase or decrease results in an increase or decrease in other capitals, but this is not a rule. One can imagine a situation when an increase or decrease in the level of one capital causes a decrease or increase in another or other capitals.

(4) The balance point between the company's capital means at the same time the maximum efficiency of the company, but because the level of individual capitals is constantly changing, the balance point reached is temporary. For this reason, sustainable capital management is an on-going process.

(5) For efficient capital management, one should know the value of individual capitals and, in the case of some capitals, e.g., social capital, both the measurable monetary value and its level (in irrational units, e.g., very high, high, low, very low).

(6) The number of capitals and qualifying individual components to it is a matter of managers. It is important that their number and allocation of components to individual capitals be maintained in the long run due to the possibility and purposefulness of comparing effects over time.

We can calculate the progress of reaching the equilibrium point in many ways. It can be expressed, e.g., by the quotient of the sum of differences between the current values of individual capitals and the optimal values of those capitals ensuring an equilibrium of capitals by the number of capitals included

in the calculations. Such measurement can be made even every day, however, such a high frequency is not needed for the day-to-day management of the enterprise. It can be expected that, in practice, the measurements would be carried out on a monthly, quarterly, and annual basis. Throughout the study, it was possible to develop a mathematical approach to the discussed concept in the form of two coefficients: the average percentage difference of capitals and weighted capital differences.

The ratio of the average percentage difference in capital may be expressed through a following formula:

$$1 - \frac{\sum_{i=1}^{6} \frac{|k_{o_i} - k_{d_i}|}{k_{o_i}}}{6} \tag{1}$$

This ratio informs us about average capital mismatch, it is very sensitive to large deviations of even one of the capitals. If it is close to 1 it means that the capitals are close to the optimal level. If it is close to 0, it means that the level of capital is significantly different from the optimal (expected) value. It should be remembered that average maladjustment means that not all, but only some of the capitals may deviate from the optimal level.

The weighted capital difference ratio has only a slightly more complicated structure:

$$1 - \frac{\sum_{i=1}^{6} \frac{k_{o_i}}{K} \cdot |k_{o_i} - k_{d_i}|}{\sum_{i=1}^{6} \frac{k_{o_i}^2}{K}} = 1 - \frac{\sum_{i=1}^{6} k_{o_i} \cdot |k_{o_i} - k_{d_i}|}{\sum_{i=1}^{6} k_{o_i}^2} \tag{2}$$

The value of this coefficient informs us about the company's effectiveness as a whole. The differences in each of the constituent capital are measured by the share of the capital in the total value of the enterprise. If it is close to 0, it means that the most significant capitals for a given entity (i.e., those that currently had the highest values) are at a very poor level. If it is close to 1, it means that changes to target values should be insignificant.

Only these two factors together inform us about the company's condition because, while the second one is good for characterizing its overall efficiency, the first one detects large errors on individual capitals, even those with the lowest value at the moment.

## 3. Results

The study was carried out in the Pamapol SA company located in Ruśc in central Poland. The company is one of the significant producers of ready meals on the Polish market. The offer includes canned meat, pates, ready soups, and canned vegetables, sold both in the traditional channel through wholesalers and in large commercial networks. The company employs approx. 400 employees. Capital values Pamapol SA in Table 1.

**Table 1.** The value of Pamapol's capitals as at 31 December 2018.

| The Share Capital | (in Thous.) PLN | % |
|---|---|---|
| fixed assets | 9175 | 40.90 |
| financial | 2157 | 9.62 |
| structural | 475 | 2.12 |
| human | 6111 | 27.24 |
| market | 3979 | 17.74 |
| social | 534 | 2.38 |
| Total | 22,431 | 100.00 |

The capital structure was compared with the structure of two leading companies on the market.

Company I is very similar to Pamapol in terms of size and the type of activity. This company is one of the largest producers of poultry products. It distributes sausages, sausages, pies, dinner dishes,

and meats in a rawer form. The company is involved in social campaigns by organizing various sports tournaments or competitions.

Company II is similar in size and also operates within the food industry, but the type of production varies. This company is one of the largest producers of dairy products. It offers pasteurized milk, butter and fats, flavored milk, soft brine cheeses, and hard ripening cheeses. The company is known for caring for the environment.

Calculation results in Tables 2–5.

**Table 2.** Calculation of differences in the capital structure between Pamapol SA and company I in PLN.

| Corporate Capitals | Symbol | Current Capital Value | Symbol | Optimal Capital Value | Formula | Difference |
|---|---|---|---|---|---|---|
| fixed assets | $k_{o1} =$ | 9,175,000 | $k_{d1} =$ | 8,448,000 | $|k_{o1} - k_{d1}| =$ | 727,000 |
| structural | $k_{o2} =$ | 2,157,000 | $k_{d2} =$ | 3,125,000 | $|k_{o2} - k_{d2}| =$ | 968,000 |
| market | $k_{o3} =$ | 475,000 | $k_{d3} =$ | 664,000 | $|k_{o3} - k_{d3}| =$ | 189,000 |
| human | $k_{o4} =$ | 6,111,000 | $k_{d4} =$ | 5,509,000 | $|k_{o4} - k_{d4}| =$ | 602,000 |
| financial | $k_{o5} =$ | 3,979,000 | $k_{d5} =$ | 2,832,000 | $|k_{o5} - k_{d5}| =$ | 1,147,000 |
| social | $k_{o6} =$ | 534,000 | $k_{d6} =$ | 1,592,000 | $|k_{o6} - k_{d6}| =$ | 1,058,000 |

**Table 3.** Calculation of differences in the capital structure between Pamapol SA and company I in PLN—continued.

| Corporate Capitals | Difference | Formula | Difference Percentage | Formula | Capital Weights |
|---|---|---|---|---|---|
| fixed assets | 727,000 | $|k_{o1} - k_{d1}|/k_{o1} =$ | 8% | $k_{o1}/K =$ | 0.4090 |
| structural | 968,000 | $|k_{o2} - k_{d2}|/k_{o2} =$ | 45% | $k_{o2}/K =$ | 0.0962 |
| market | 189,000 | $|k_{o3} - k_{d3}|/k_{o3} =$ | 40% | $k_{o3}/K =$ | 0.0212 |
| human | 602,000 | $|k_{o4} - k_{d4}|/k_{o4} =$ | 10% | $k_{o4}/K =$ | 0.2724 |
| financial | 1,147,000 | $|k_{o5} - k_{d5}|/k_{o5} =$ | 29% | $k_{o5}/K =$ | 0.1774 |
| social | 1,058,000 | $|k_{o6} - k_{d6}|/k_{o6} =$ | 198% | $k_{o6}/K =$ | 0.0238 |

**Table 4.** Calculation of differences in the capital structure between Pamapol SA and company II in PLN.

| Corporate Capitals | Symbol | Current Capital Value | Symbol | Optimal Capital Value | Formula | Difference |
|---|---|---|---|---|---|---|
| fixed assets | $k_{o1} =$ | 12,455,000 | $k_{d1} =$ | 12,674,000 | $|k_{o1} - k_{d1}| =$ | 219,000 |
| structural | $k_{o2} =$ | 2,480,000 | $k_{d2} =$ | 3,056,000 | $|k_{o2} - k_{d2}| =$ | 576,000 |
| market | $k_{o3} =$ | 546,000 | $k_{d3} =$ | 1,100,000 | $|k_{o3} - k_{d3}| =$ | 554,000 |
| human | $k_{o4} =$ | 6,512,000 | $k_{d4} =$ | 6,780,000 | $|k_{o4} - k_{d4}| =$ | 268,000 |
| financial | $k_{o5} =$ | 4,309,000 | $k_{d5} =$ | 4,551,000 | $|k_{o5} - k_{d5}| =$ | 242,000 |
| social | $k_{o6} =$ | 540,000 | $k_{d6} =$ | 752,000 | $|k_{o6} - k_{d6}| =$ | 212,000 |

**Table 5.** Calculation of differences in the capital structure between Pamapol SA and company II in PLN—continued.

| Corporate Capitals | Difference | Formula | Difference Percentage | Formula | Capital Weights |
|---|---|---|---|---|---|
| fixed assets | 219,000 | $|k_{o1} - k_{d1}|/k_{o1} =$ | 2% | $k_{o1}/K =$ | 0.4640 |
| structural | 576,000 | $|k_{o2} - k_{d2}|/k_{o2} =$ | 23% | $k_{o2}/K =$ | 0.0924 |
| market | 554,000 | $|k_{o3} - k_{d3}|/k_{o3} =$ | 101% | $k_{o3}/K =$ | 0.0234 |
| human | 268,000 | $|k_{o4} - k_{d4}|/k_{o4} =$ | 4% | $k_{o4}/K =$ | 0.2426 |
| financial | 242,000 | $|k_{o5} - k_{d5}|/k_{o5} =$ | 6% | $k_{o5}/K =$ | 0.1605 |
| social | 212,000 | $|k_{o6} - k_{d6}|/k_{o6} =$ | 39% | $k_{o6}/K =$ | 0.0201 |

With such amounts of capital, the ratio of average percentage differences is 0.45, while the weighted capital difference ratio is 0.88.

With such amounts of capital, the ratio of average percentage differences is 0.71, while the weighted capital difference ratio is 0.97.

Calculations show significant differences between the current capital structure and the capital structure of two leading companies in the industry. The largest ones occur mostly in the range of

market and social capitals. The analyzed case study does not allow one to make general conclusions, however, it shows that calculations of efficiency, in this way, are possible, but the real problem comes from the correct valuation of companies' capital.

Generally, performed calculations have shown that new indicators allow one to measure the effectiveness of an enterprise in practice. However, a number of comments and conclusions appear at this stage of the research.

## 4. Discussion

The key issue in the new concept is the correct valuation of the company's capital, both current and optimal. There are many methodological problems in both of these valuations. At the same time, for capital measurement to be widely used by companies, it must be based on very simple principles. The simplest methods are comparative and cost-based. However, in comparative methods you never know whether the pattern with which you compare is optimal, and in cost-based methods the scale of expenditure does not always mean that they are effective and will yield the desired effect. There are undoubtedly a number of problems that should be the subject of further search for the best solutions.

Looking for the value of current capital, cost methods seem to be the best, although, as mentioned above, expenditure does not always mean a good effect. The most significant problems are as follows:

a   Fixed assets. For the valuation of fixed assets (movables), instead of the demand and supply ratio, an asset's usefulness ratio can be used to generate revenue. The assessment of the technical wear of machines and devices can be calculated from the formula: the current service life/the assumed period of technical life under specific operating conditions. The assessment of the degree of functional wear and tear was calculated in accordance with the formula: device element's service length/length of the device's functional lifespan. Optionally, for some fixed assets, their value was calculated as: financial outlays on bringing the device to full functional usefulness, e.g., the replacement of parts or repair/cost of a new element. For land valuation, average land prices published by the statistical office can be used. For the valuation of buildings, it is best to use the income-based method. The value of the property is then calculated from the formula: net operating income accepted for capitalization x 1/capitalization rate.

b   Financial capital. The basis of the measurement was the verification of current assets, e.g., in their reduction by receivables and unfit inventories. The valuation should also consider whether to include the shares held in other companies. The valuation is subject to the value of the enterprise as a profit-generating entity and not the value of the company to shareholders. Therefore, if the shares or shares do not bring dividends, their value must be neglected.

c   Human capital. For the calculation of the value of capital, a simple cost method can be used, in which the value of human capital was calculated as the product of the number of employees and the value of remuneration during the period of gaining the professional experience necessary to perform work in individual positions. In addition, these values increase other labour costs, e.g., employee recruitment, studies, courses, training.

d   Market capital. The easiest way to use a simple cost method, where the value of this capital is determined by the expenditure on advertising, fairs, exhibitions, promotional campaigns, and the development of the sales network. In addition, if there is significant brand value, it should be accompanied by its valuation, e.g., using the discounted license fees method.

e   Structural capital is the sum of expenditures on research and development, the purchase of licenses and patents, computers and software, etc. Similar to the previous cases, the cost-based method was used.

f   Social capital is the difference between the total value of capitals (value of the enterprise) and the following capitals: fixed assets, financial, human resources, market, and structural.

As to searching for optimal capital values, three approaches can be used:

a　　Comparing the current capital structure of the surveyed enterprise with the structure of an industry-leading enterprise (or two-three) having not only a recognized brand but having achievements in the field of positive impact on the natural environment and achievements in the field of corporate social responsibility. This enterprise should be similar to the respondent not only in size but also in the type of activity;

b　　Comparing the current capital structure of the audited enterprise with the capital structure in the industry (several to several dozen companies). Due to a larger number of companies, the reliability of capital structure data increases, but there are two problems. First of all, not all companies in a given industry may be leading in relation to the examined enterprise, and, secondly, there may be companies in the industry that are significantly different in various respects.

c　　Searching for optimal capital within the examined enterprise without comparing with other enterprises. Here, the principles described above can be used partly to value current capital, paying attention to its level. In the valuation of social capital, many problems may be noticed. However, if we examine the level of social capital (atmosphere at work, trust, and relationships between employees and between employees and managers) using surveys and it will be, e.g., 53% (the sum of employee responses that indicate a very good and good level of capital), and the optimal value will be 100% positive answers. If we know the current value of this capital in monetary values, we know the value of optimal capital.

## 5. Conclusions

The strengths of the presented idea are as follows:

(1)　It enables the replacement of a profit indicator that does not match today's modern economic and social realities for the needs of effective enterprise management. It is important that the concept does not assume the liquidation of profit as a measure, however limiting its role to the issue of tax settlements.

(2)　Combining the assessment of the work of management boards with the assessment of the economic condition of the enterprise. In the case of profit, it is not entirely clear whether profit is a measure of the economic condition of an entity or a measure of the ability and effort of a manager and employees.

(3)　The current level of knowledge about capitals, which allows one to measure not only their level in irrational units but also in monetary values. In particular, the existence of fairly simple valuation methods (cost-based methods) is important.

(4)　The high flexibility of rules regarding the selection of the number and type of capitals, and the period and methods of measuring the value of individual capitals. Flexibility in terms of rules is not in contradiction with the possibility of comparing levels of capital efficiency ratios not only in a given industry but in the entire economy.

(5)　Ensuring balance between commercial and social goals in the enterprise. This is because social (external and internal) as well as market and structural capital must be adapted to the level of other capital. This eliminates the age-old dilemma of managers of public utilities: profit or the best satisfaction of the needs of the population and entrepreneurs.

The weaknesses of the idea include:

(1)　A lack of research on a larger scale to assess the concept, especially in enterprises with an unstable economic situation and in various industries.

(2)　A lack of wider knowledge about the mutual influence of capital on each other. It is likely that this impact is conditioned by a number of different factors and it will not be possible to build a uniform and universal rule in this respect for all enterprises for the needs of management, except for general guiding principles.

(3)     A lack of a built-in cost accounting system for capital valuation purposes. In the study described in the article, most of the cost-related data was prepared manually by employees, which makes it impossible to quickly assess effectiveness.

(4)     The need to apply the same capital valuation principles in individual years for the purpose of comparability of results. This can be difficult, especially in the first periods after the first experiments, where comments and the legitimacy of the modification of assumptions will certainly appear.

(5)     The views and long-standing habits as well as a resistance to new things and changes among the employees of enterprises and institutions regarding the role of net profit as a universal measure.

**Funding:** This research received no external funding.

**Acknowledgments:** I am grateful to the anonymous reviewers for excellent comments on an earlier drafts.

**Conflicts of Interest:** The author declares no conflict of interest.

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
