# Peer review of "Sustainable Enterprise Capital Management"

_economies, doi:10.3390/economies8010012_

Round 1

Reviewer 1 Report

Dear Author / Authors,

Please find attached the Review Report Form for your Article: Title: “Sustainable enterprise capital management”, Journal: Economies (ISSN 2227-7099), Manuscript ID: economies-685760, Type: Article, Number of Pages: 11, Submission Date: 18 December 2019.

Congratulations for your study and for your hard work!

Kind regards,

The Reviewer

Author Response

Thank you very much for the extensive review. It is very important to me. I have also read carefully two recommended publications. I cited both publications in my article. I will use them even more in my subsequent publications on sustainable corporate capital management. Thank you once again for the review and comments.

Best regards,

Reviewer 2 Report

In my opinion the study meet the requirements of the learned paper.

I can not find an inappropriate part, transparent, logically structured paper.

The literature is also excellent and comprehensive to the topic.

I recommend the publishing. 

Author Response

Thank you very much for your review. The author rarely receives such positive reviews and it is very important to me. Thank you once again.

Best regards,

Reviewer 3 Report

My general comments: 

Introduction part should emphasise, why this paper/study is relevant and to whom? 

I do not find the main research question of this paper/study. 

More stronger Methodological part/ position of paper should be demonstrated: authors should see to answer key research questions about the origin of knowledge and the validity of knowledge.

Answering the main research question will be added value of this paper/study.

Please provide more details on development your research framework.

Author Response

Thank you very much for the extensive review. It is very important to me. I also carefully read the presented comments.

Regards,

Note 1 and answer:

Introduction part should emphasize, why this paper/study is relevant and to whom? 

In the answer to the question, the following fragment was included in the text at the end of the introduction:

"The article is part of the search for an alternative to profit as the current basic efficiency measure in the world. It is important primarily for company managers and owners who can better use the company's capital and achieved better business results, also as a result of increasing revenues or reducing costs. But it should also be emphasized that this way of measuring efficiency pays special attention to human and social capital, often underestimated when using profit as a measure of effectiveness. In addition, it includes elements of corporate social responsibility in the assessment, combining public utilities with commercialism. "

Note 2 and answer:

I do not find the main research question of this paper/study.

In the first paragraph of the Materials and methods chapter the content was supplemented with a basic research question. Added text "During research, the author tries to answer the research question - do the indicators developed by the author measure the effectiveness of the enterprise in practice?"

Note 3 and answer:

More stronger Methodological part/ position of paper should be demonstrated: authors should see to answer key research questions about the origin of knowledge and the validity of knowledge

I have a problem answering this comment. The presented concept derives primarily from two theories: resources and intellectual capital. I write quite a lot about this in the introduction. The concept of a new way of measuring efficiency by means of a balance point between capital is new, not previously described in the literature. It has an original character and it is difficult to say how much knowledge about it.

Note 4 and answer:

Answering the main research question will be added value of this paper/study.

The response to the attention is positive. At the end of the chapter Results of research, the following was added: "Generally performed calculations have shown that new indicators allow to measure the effectiveness of an enterprise in practice. However, a number of comments and conclusions appear at this stage of the research. " 

Note 5 and answer:

Please provide more details on development your research framework.

It must be taken into account that in order to identify any regularities, further research on a large group of enterprises is necessary. Only pilot studies are described in the article. The article only proposes a new concept of efficiency calculations and is a signal and an incentive for further research verifying the concept. The development of the methodological part of the research is already being carried out by the author, followed by empirical research on a group of about 20 enterprises by the end of 2020. 

Round 2

Reviewer 1 Report

I have received the revised version of your Article entitled “Sustainable enterprise capital management” (Journal Economies (ISSN 2227-7099), Manuscript ID economies-685760).

Thank you kindly for your message concerning your Article as well as the description of the steps that you took into consideration in order to prepare the final version for the Journal Economies (ISSN 2227-7099), MDPI. 

After carefully reading the improved version of your Manuscript I would like to congratulate you for a work well done and also, to wish you good luck with the publication of your current as well as your future papers!

I wish you well!

Kind regards,

The Reviewer